# Synthesis and Application of Sol-Gel-Derived Nano-Silica in Glass Ionomer Cement for Dental Cementation

**DOI:** 10.3390/biomimetics10040235

**Published:** 2025-04-10

**Authors:** Mohammad Dharma Utama, Nina Ariani, Edy Machmud, Acing Habibie Mude, Muhammad Akira Takashi Dharma, Aksani Taqwim, Risnawati Risnawati

**Affiliations:** 1Department of Prosthodontics, Faculty of Dentistry, Hasanuddin University, Perintis Kemerdekaan KM. 10 Street, Makassar 90245, Indonesia; machmudedy@gmail.com (E.M.); acinghabibie@unhas.ac.id (A.H.M.); risnawati.risna84@gmail.com (R.R.); 2Department of Prosthodontics, Faculty of Dentistry, Indonesia University, 4 Salemba Raya Street, Jakarta Pusat 10430, Indonesia; nina.ariani02@ui.ac.id; 3Department of Dental Material, Faculty of Dentistry, Hasanuddin University, Perintis Kemerdekaan KM. 10 Street, Makassar 90245, Indonesia; 4Department of Prosthodontics, Faculty of Dentistry, Muslim Indonesia University, 27 Pajonga Dg. Ngalle Street, Makassar 90122, Indonesia; aksani.taqwim@umi.ac.id

**Keywords:** luting materials, nano-silica, *Thalassiosira* sp., layer thickness, surface roughness, compressive strength, tensile strength

## Abstract

Although glass ionomer cements (GIC) are widely used in dental restorations, their long-term performance remains limited by their mechanical properties, including surface roughness and fracture resistance. This study investigates the synthesis of nano-silica from *Thalassiosira* sp. diatoms through the sol-gel process and its application in influencing the mechanical and physical properties of GIC luting materials. A control group and three experimental groups of different nano-silica concentrations (1%, 3%, and 5%) were compared. Several analyses, including confocal laser scanning microscopy (CLSM), scanning electron microscopy (SEM), and universal testing machines (UTM), were used to determine layer thickness, surface roughness, compressive strength, and tensile strength. Statistical analysis exhibited significant differences between the groups (*p* < 0.05). The 3% nano-silica group indicated an optimal compromise between mechanical strength and surface smoothness, while the 5% group showed increased thickness and roughness with slightly lower strength. These findings emphasize that the sol-gel-derived nano-silica from *Thalassiosira* sp. potentially enhances certain characteristics of GIC for possible dental cementation. Further research is needed to determine the long-term durability and bioactivity of these modified materials.

## 1. Introduction

The use of glass ionomer cement (GIC) in luting procedures like dental bridge cementation has made it a vital component of restorative dentistry. GICs are a well-known option in contemporary dentistry practice because of their clinical handling features, fluoride-releasing capabilities, and biocompatibility [1,2]. Nevertheless, most GIC materials have substantial limitations despite these benefits. The need to reinforce important characteristics, particularly their inherent brittleness and low fracture resistance, has long been recognized by clinicians and researchers [3]. The ability of dental restorations to withstand the functional stresses produced during mastication and other oral pressures depends on these mechanical and physical characteristics [4].

Currently used in clinical practice, GIC often exhibits inadequate mechanical strength, compromising the longevity of dental restorations [3,5]. Moreover, it is still quite difficult to achieve a smooth surface finish, which has an impact on both resistance to plaque resistance and aesthetic outcomes [6]. The incorporation of additional components can increase the thickness of the cement layer, despite several studies focusing on developing modified GIC formulations to address these limitations while maintaining the material’s fundamental advantages [7,8]. Such a shift in dimensions could lead the restoration to be positioned incorrectly, which would compromise the occlusal and functional relationships.

To overcome these limitations, various nanoparticle reinforcements such as nano-hydroxyapatite, -zirconia, -titania, -alumina, and -silica have been explored [9,10,11]. Among various nanomaterials, nano-silica has emerged as a promising additive for GIC reinforcement due to its ability to improve cross-linking and promote denser matrix development within the GIC structure [12]. Furthermore, the inclusion of nano-silica in GIC formulations has shown increased surface smoothness [13]. These alterations contribute to significant increases in the physical and mechanical properties of GICs.

The sol-gel method is a widely used technique for synthesizing nanostructures, including silica, through hydrolysis, condensation, gelation, aging, drying, solidification, and crystallization processes. This method was chosen for nano-silica production due to its precise material control over particle size, and morphology, facilitating the production of nanoparticles ideal for homogeneous integration into dental materials [14]. Previous studies of sol-gel-derived nano-silica have demonstrated promising outcomes in enhancing the properties of GIC, including mechanical strength, biocompatibility, and bioactivity, which is a potential material for dental restorations [15,16,17].

In the present study, silica was extracted from the marine diatom *Thalassiosira* sp., an abundant natural source rich in biogenic silica. The diatoms were subjected to chemical treatment, calcination, and sol-gel synthesis to yield amorphous nano-silica. Importantly, the structural morphology of the diatom frustules was not preserved nor was it the focus of this study. Instead, the diatom source was selected for its ecological abundance and high silica content, offering a sustainable and cost-effective alternative for producing nano-silica. The research does not investigate the architectural influence of the diatom skeleton; rather, it evaluates the role of sol-gel-derived nano-silica particles in enhancing the performance of GIC materials.

The objective of this study is to investigate the mechanical and physical characteristics of GIC luting materials affected by the addition sol-gel-derived nano-silica from *Thalassiosira* sp., examining layer thickness, surface roughness, compressive strength, and tensile strength. The outcome of this research is expected to have direct clinical relevance by potentially improving mechanical resistance. Moreover, this study aims to support future investigations into the long-term biocompatibility and scalability of nano-silica integration in GIC formulation for broader clinical use.

## 2. Materials and Methods

### 2.1. Sol-Gel and Nanomateial Processing

A total of 500 g of *Thalassiosira* sp. diatoms were cleaned and sun-dried. The dried diatoms were then treated with 1 M HCl for 1 h to remove impurities, rinsed with distilled water, and dried again. The purified diatoms were calcined in a furnace at 600 °C for 7 h to obtain silica-rich ash. At this stage, the original structure of the diatom frustules was deliberately removed through high-temperature calcination. The purpose of this process was to obtain purified biogenic silica from the diatoms, which would then be used for nano-silica synthesis. There was no intention to preserve or investigate the natural architecture of the diatom skeletons. As a result, the structural features of the diatoms were completely eliminated and did not play any role in the formation of the silica hydrogel or the final nano-silica material.

A 20 g portion of the resulting ash was mixed with 500 mL of 1 M NaOH and stirred at 60–80 °C for 3 h to produce a sodium silicate solution. The solution was filtered to remove any residual solids, yielding a clear sodium silicate filtrate.

The sodium silicate solution was then titrated with 1 M HCl dropwise while continuously stirring until the pH reached 7. During this titration process, the reaction temperature was consistently maintained between 60 °C and 80 °C, and the stirring speed was precisely controlled using a magnetic stirrer. This process led to the formation of a silica hydrogel, which was allowed to stand overnight to complete gelation. The resulting hydrogel was washed with distilled water until the washing water reached a neutral pH. The purified hydrogel was then oven-dried at 80 °C to form a silica xerogel.

The silica xerogel was ground and sieved to obtain a fine powder, which was subsequently analyzed using Fourier-transform infrared spectroscopy (FT-IR) (IRXross, Shimadzu Corporation, Kyoto, Japan) to confirm presence of silica. The FT-IR was used not only to confirm the presence of silica but also to identify specific functional groups such as siloxane (Si–O–Si), silanol (Si–OH), and hydroxyl (–OH) groups. These surface functional groups are important indicators of successful sol-gel synthesis and are relevant to the potential interaction between nano-silica and the polyacid component of GIC, contributing to improved particle dispersion and matrix bonding.

The final step involved converting the silica xerogel into nano-silica using an ultrasonic milling method. Ultrasonic waves, with frequencies ranging from 20 kHz to 10 MHz, were applied to the silica powder in a liquid medium. This process utilizes the principle of acoustic cavitation, where sound vibrations create microscopic bubbles in the liquid, leading to the breakdown of silica particles into nanoscale dimensions. The resulting nano-silica particles were characterized by their uniform size and morphology, making them suitable for incorporation into glass ionomer cement (GIC).

### 2.2. Preparation the Specimen

After obtaining the nano-silica, the specimens were prepared according to the study groups. This experimental study tested the cylindrical specimens composed of Glass Ionomer Cement (GIC) Type 1 (GC Fuji I, GC Corporation, Tokyo, Japan), divided into four study groups in Table 1.

The materials for Group 1 and the other groups were mixed following the manufacturer’s instructions, using the GIC liquid and the power-to-liquid ratio for 15 s. The mixed materials were placed in custom acrylic molds (dimensions: 4 × 6 mm), covered with celluloid strips, and compressed with a 0.5 kg load to ensure uniform density. Once the matrices were removed, the homogenized specimens were taken out of the molds and stored in an incubator at 37 °C for 24 h. After all specimens were completed, Scanning Electron Microscopy (SEM) (SU3800, Hitachi High-Tech, Tokyo, Japan) was investigated to examine the morphology in each group.

### 2.3. Physical Characteristic Test

To measure surface thickness and roughness, the specimens were first subjected to a static pressure of 150 N in the molds for five minutes. The thickness was measured in the gap between the two acrylic mold parts using a Confocal Laser Scanning Microscope (CLSM) (Olympus OLS 4000 LEXT, Tokyo, Japan) with a 100-lens and 20× magnification. Subsequently, the roughness was assessed on a flat surface using a CLSM with a 100-lens and 20× magnification. Surface roughness (Ra) was recorded in micrometers (µm), where lower values indicate smoother surfaces. A smoother surface is generally preferred in dental restorations due to reduced plaque accumulation and improved aesthetics.

### 2.4. Compressive Strength Measurement

Compressive strength was measured per ISO 9917-1 using a Universal Testing Machine (UTM) (MTS Bionix Tabletop, Eden Prairie, MN, USA). The crosshead speed was set at 0.005 mm/s, and compressive stress was calculated using(1)$$\delta_{max}=\frac{{Ν}_{max}}{A}$$
where δmax is the maximum compressive stress [MPa], Νmax is the maximum compressive load [Ν], and A is the cross-sectional area [mm^2^].

### 2.5. Tensile Strength Evaluation

Tensile strength also was evaluated using UTM with the Diametral Tensile Strength (DTS), applying tensile stress through compression:(2)$$DTS=\frac{2Ρ}{\left(\pi DT\right)}$$
where P is the applied load, D is the cylinder diameter, and T is the cylinder thickness. DTS values [kgf/cm^2^] were converted to MPa using the formula DTS [MPa] = DTS [kgf/cm^2^] × 0.09807.

### 2.6. Statistic Analysis

After completing the tests, data were collected for each variable and analyzed using SPSS software (SPSS Statistics 26.0, New York, NY, USA). The normality of the data was assessed with the Shapiro–Wilk test, and homogeneity was evaluated using Levene’s Test to meet the assumptions for one-way ANOVA testing. Comparisons between groups were made, followed by post hoc analysis using Tukey’s HSD test with a significance level set at *p* < 0.05.

## 3. Results

### 3.1. Fourier-Transform Infrared Spectroscopy (FT-IR)

The utilization of Fourier-transform infrared spectroscopy (FT-IR) for the characterization of silica obtained from the diatom *Thalassiosira* sp. through sol-gel and sonication techniques seeks to detect the presence of siloxane (Si-O-Si), silanol (Si-OH), and siloxyl (Si-O) functional groups. The FT-IR spectral data in Figure 1 indicates that the synthesized silica from *Thalassiosira* sp. exhibits a wavenumber range of 4000–500 cm^−1^. The most prominent peak associated with the siloxane bond, particularly the asymmetric stretching of Si-O-Si, is detected at a wavenumber of 1050 cm^−1^, whereas the silanol (Si-OH) peak is found at 850 cm^−1^, and the siloxyl (Si-O) peak is situated at 800 cm^−1^. Furthermore, a hydroxyl (-OH) stretching functional group is detected at a wavenumber of 3450 cm^−1^. The presence of these functional groups, particularly Si–OH and –OH, confirms the successful formation of surface-active sites on the synthesized silica. These groups are known to be beneficial for subsequent chemical interactions in material formulations such as GIC.

### 3.2. Scanning Electron Microscopy (SEM)

The characterization of the nano-silica obtained from diatom *Thalassiosira* sp. at concentrations of 1%, 3%, and 5% when mixed with Luting Glass Ionomer Cement was conducted using scanning electron microscopy (SEM) to identify the surface morphology of the nano-silica in combination with the cement. As shown in Figure 2, the crystalline structure of the Luting Glass Ionomer Cement (without the addition of nano-silica) is predominant over its lanthanide structure. In contrast, the mixtures containing 1%, 3%, and 5% nano-silica exhibit an amorphous structure of nano-silica derived from the diatom *Thalassiosira* sp. (indicated by the red arrows), interspersed with the crystalline structure of the Luting Glass Ionomer Cement (indicated by the black arrows). Notably, agglomeration occurs among the amorphous nano-silica structures, with some nano-silica particles positioned on the surface of the Luting Glass Ionomer Cement, while others are located at its edges.

### 3.3. Physical and Mechanical Findings

The results of the one-way ANOVA test demonstrate significant differences among groups for all tested variables (thickness, roughness, compressive strength, and tensile strength), with *p* < 0.001. The TukeyHSD post hoc test revealed differences among groups, denoted by unique superscript letters for each variable, as shown in Figure 3.

### 3.4. Layer Thickness

The mean thickness (Table 2) values across the various groups, as measured by CLSM, ranged from 20 to 28 µm, with significant differences identified (*p* < 0.001). The NANO-SILICA5% group demonstrated the greatest thickness, measuring 28.43 µm, which is significantly higher than that of all other groups. The NANO-SILICA1% group exhibited the lowest mean thickness at 20.27 µm, a statistically significant difference compared to all other groups. The CONTROL and NANO-SILICA3% groups exhibited intermediate levels, with values approximately 24 µm and 22 µm, respectively.

### 3.5. Surface Roughness

In Table 3, there was a significant difference in surface roughness between the groups (*p* < 0.001). The CONTROL and NANO-SILICA5% groups exhibited the highest roughness values, recorded at 0.19 µm, suggesting a statistically comparable surface texture. The NANO-SILICA1% group demonstrated a reduced roughness value (0.151 ± 0.007 µm) in comparison to the previously mentioned groups. According to CLSM surface analysis, the NANO-SILICA3% group exhibited the lowest surface roughness at 0.128 µm, showing a statistically significant decrease in roughness relative to all other groups.

### 3.6. Compressive Strength

The compressive strength values among the various experimental groups exhibited notable variability across the materials, as illustrated in the Table 4. CONTROL demonstrated the highest compressive strength at 78.8 MPa, significantly surpassing all other groups. The post hoc test indicated that the compressive strength of the NANO-SILICA groups varied by concentration, with mean values of 64.6 MPa for NANO-SILICA3%, 48.0 MPa for NANO-SILICA5%, and 31.8 MPa for NANO-SILICA1%.

### 3.7. Tensile Strength

The tensile strength analysis indicated significant differences among the groups (*p* < 0.001, ANOVA one-way test), with post hoc results reflecting trends similar to those found in the compressive strength data. CONTROL exhibited the highest tensile strength at 1.32 MPa, with NANO-SILICA3% following closely at 1.24 MPa. Tukey’s HSD test indicated no statistically significant difference between the two groups. NANO-SILICA1% and NANO-SILICA5% exhibited notably reduced tensile strengths of 0.74 MPa and 0.76 MPa, respectively, which were statistically distinct from those of CONTROL and NANO-SILICA3% (Table 5).

## 4. Discussion

The incorporation of nano-silica particles derived from *Thalassiosira* sp. diatom extract presents a promising approach to improving the physical and mechanical properties of Glass Ionomer Cement (GIC). The results of this investigation indicated that the layer thickness, surface roughness, compressive strength, and tensile strength of the GIC were significantly influenced by the concentrations of nano-silica.

The findings indicate that the thickness of the GIC layer increased with the incorporation of nano-silica, particularly at higher concentrations. The formulation with 5% nano-silica demonstrated the highest thickness at 28.43 µm. This increase can be primarily attributed to nanoparticle agglomeration, a common occurrence at higher filler loadings, which has the potential to trigger an uneven distribution within the GIC matrix. Previous research has discovered that the combination activity of nanoparticles mixed with materials indicates that increased nanoparticle concentrations can result in particle clustering, thus raising material thickness [18]. Moreover, SEM analysis confirmed the presence of these agglomerations, visually demonstrating significant particle clustering within the cement matrix at higher nano-silica concentrations, thereby providing direct microstructural evidence of uneven particle distribution that contributes to the increased thickness.

In contrast, adding 1% nanoparticles exhibited the lowest mean thickness of approximately 20.27 µm. The lower concentration of nano-silica likely contributed to a more uniform dispersion of nanoparticles within the GIC matrix, potentially supporting a consistent and thinner cement layer. These observations are consistent with previous research indicating that lower nanoparticle concentrations tend to facilitate improved particle dispersion, resulting in more uniform structural properties in dental restorative materials [19]. SEM observations from this study further support these results, showing considerably fewer clusters and a visibly more homogeneous distribution of nanoparticles within the cement at the 1% concentration, thus validating the hypothesis of improved uniformity at lower nanoparticle loadings.

The intermediate results for the control group (24.11 µm) and 3% of nano-silica (22.53 µm) suggest that an optimal concentration range may exist for incorporating nano-silica into GIC. This intermediate range potentially offers an optimal balance between enhanced physical and mechanical properties from nano-silica addition and maintaining uniform dispersion to prevent excessive thickness or structural irregularities. At moderate concentrations, nanoparticles likely achieve sufficient dispersion to avoid significant agglomeration while providing improved cross-linking and reinforcing effects within the cement matrix [20]. Determining such an optimal concentration is critical for maximizing clinical performance, ensuring the cement layer is sufficiently thin and uniform to maintain proper fit, marginal integrity, and mechanical reliability of dental restorations.

A smoother surface is desirable in dental applications, as it minimizes bacterial adhesion and plaque accumulation, which are critical factors for long-term dental restoration success [21]. In this study, surface roughness values varied significantly depending on the nano-silica concentration added to the glass ionomer cement (GIC). The smoothest surface observed in the group containing 3% nano-silica (0.128 µm) suggests that this particular concentration optimally balances nano-silica dispersion within the GIC matrix, yielding a homogeneously refined surface texture. Although roughness was the parameter measured, the term ’smoothness’ is used in this section to describe surfaces with lower roughness values, which are clinically favorable.

Previous research suggests a threshold surface roughness of 0.2 µm, below which no further reduction in bacterial accumulation is observed [18]. Importantly, all groups in this investigation recorded surface roughness measurements below this critical threshold, indicating their suitability in minimizing bacterial adherence clinically.

The reduction in surface roughness in the 3% nano-silica group may be explained by the improved nanoparticle distribution and the resulting more compact matrix structure provided by this moderate concentration. Adequate dispersion of nano-silica particles may contribute to filling microscopic surface voids or irregularities inherent in the GIC matrix, consequently creating a more uniform and smoother surface texture. This finding aligns with previous studies, emphasizing the beneficial role of nano-silica in enhancing the material’s surface characteristics by promoting denser packing and reducing surface porosity [13].

Conversely, the 5% nano-silica group exhibited increased surface roughness (0.188 µm), which is significantly higher than that of the 3% group. The elevated roughness values at higher nano-silica concentrations are likely due to nanoparticle agglomeration, as confirmed by scanning electron microscopy (SEM) observations in this study. Agglomeration can result in localized irregularities and rougher surface topography, adversely affecting the material’s overall surface homogeneity. Previous research supports this interpretation, indicating that excessive nanoparticle incorporation can lead to heterogeneous dispersion and increased surface roughness, negatively impacting mechanical properties and clinical outcomes [22].

Clinically, the findings from this study suggest that incorporating nano-silica at a controlled concentration of 3% provides significant advantages in terms of surface characteristics, potentially contributing to improved longevity and clinical success of dental restorations. Further research investigating long-term clinical behavior, biofilm formation dynamics, and the biological interaction at the nano-silica-GIC interface would provide additional insights into the optimal application of these modified materials.

Compressive strength is essential for evaluating the structural integrity of dental luting materials, influencing their clinical durability under functional stresses [23]. In this study, the control group, which contained no nano-silica, exhibited the highest compressive strength (78.8 MPa), substantially outperforming all nano-silica-enhanced groups. This finding is particularly notable as it contradicts several previous studies where the incorporation of nanoparticles generally resulted in improved compressive strengths of GIC [17,24].

Interestingly, the 3% nano-silica group demonstrated a compressive strength of 64.4 MPa, which, although lower than the control, still represents a considerable strength level. This intermediate strength suggests that moderate incorporation of nano-silica does not severely compromise structural integrity, offering a practical balance between improved physical characteristics and adequate mechanical performance. Such outcomes could be due to optimal particle distribution, enhancing the matrix structure without introducing significant internal stresses or defects [25]. However, the marked reduction observed at 5% nano-silica concentration (48.0 MPa) aligns partially with previous research findings, where excessive nano-filler concentrations often result in agglomeration and poor dispersion, thereby weakening the material structure [26]. SEM analysis confirmed nanoparticle agglomeration in this group, potentially explaining the compromised mechanical properties due to increased stress concentrations and internal defects. Contrary to earlier studies reporting a positive correlation between nanoparticle concentration and mechanical strength [17,24], this study highlights a critical threshold beyond which nanoparticle reinforcement adversely affects mechanical performance.

These contrasting results underline the complexity of nanoparticle integration into dental materials and highlight the necessity for further investigations into optimal filler levels, dispersion techniques, and surface treatments to fully exploit the benefits of nano-silica without compromising the compressive strength of GICs.

The tensile strength results notably indicated that the optimal nano-silica concentration might be around 3%, as evidenced by comparable tensile strengths observed in both the control group (1.32 MPa) and the 3% nano-silica group (1.24 MPa). The similarity between these two groups suggests that the addition of nano-silica at moderate concentrations can potentially reinforce the GIC matrix without significantly compromising its inherent mechanical integrity. At this intermediate concentration, nano-silica particles appear to be effectively dispersed throughout the matrix, offering reinforcement benefits through improved cross-linking and structural stability without substantial particle agglomeration [27].

Conversely, the tensile strength for the group with a higher nano-silica concentration (5%) exhibited a noticeable reduction, with values significantly lower (approximately 0.76 MPa) compared to the control and 3% groups. Such reductions in tensile performance at higher nanoparticle loadings might be attributed primarily to the agglomeration of nano-silica particles within the GIC matrix. Agglomeration often creates localized structural weaknesses and internal stress points, potentially decreasing the overall mechanical integrity of the cement [24]. This finding contrasts with some earlier research suggesting improvements in mechanical properties at higher nanoparticle concentrations; however, the significantly higher concentrations utilized in this study likely amplified agglomeration effects, as evidenced by SEM analyses [17,27,28]. While nano-silica theoretically improves the tensile properties of dental cements, excessive particle concentrations might negate these potential benefits by impairing structural uniformity.

Similarly, the lowest nano-silica concentration group (1%) also displayed significantly reduced tensile strength, approximately 0.74 MPa, which was comparable to the 5% concentration group. This result suggests that extremely low nano-silica concentrations may not sufficiently reinforce the cement matrix, providing minimal improvement in cross-linking density and mechanical properties. At low particle loadings, there may not be enough nanoparticles to meaningfully influence the cement’s structural integrity, resulting in limited enhancement of tensile strength. While improved particle dispersion can occur at lower concentrations, these advantages seem inadequate to provide substantial mechanical reinforcement [29]. Thus, although low nanoparticle loadings reduce agglomeration, achieving a meaningful improvement in tensile properties requires an adequate minimum concentration. These findings highlight the complexity involved in determining effective nano-silica loadings for GIC reinforcement, emphasizing the necessity for carefully balanced nanoparticle concentrations to achieve desirable mechanical outcomes.

Furthermore, this study aligns with biomimetic principles by utilizing a marine biological source (*Thalassiosira* sp.) as a template for producing functional nanomaterials. The sol-gel-derived nano-silica mimics the mineral reinforcement found in natural hard tissues, such as enamel and dentin, by enhancing the mechanical stability and surface texture of restorative materials. Although the structural morphology of the diatoms was not preserved, their use as a silica source reflects a sustainable, bio-inspired strategy for material enhancement. This approach not only supports ecological material sourcing but also advances the development of dental materials that emulate nature’s strategies for durability and integration.

## 5. Conclusions

In clinical practice, the mechanical properties of luting agents are crucial for the longevity of dental restorations. The results of this study suggest that adding nano-silica at a concentration of 3% could potentially improve the performance of GIC in dental applications by enhancing surface smoothness while maintaining adequate tensile strength. However, care must be taken not to exceed this concentration, as higher amounts of nano-silica could compromise the material’s compressive strength and increase surface roughness.

The authors selected a small sample size because the prototype material used was being evaluated for the first time as an addition to glass ionomer cement. Therefore, the initial aim was to acquire baseline evidence on the use of this biomaterial in experimental design. Investigating the effects of this natural resource on the bioactivity of GIC, such as its antibacterial properties, would also be a valuable area for future research.

Within the limitations of this study, it can be concluded that the findings underscore the significant impact of varying concentrations of nano-silica on the physical and mechanical properties of the tested materials. The results provide valuable insights into the optimization of material properties through the strategic use of nano-silica, which could have significant implications for various industrial applications. Further research is warranted to explore the underlying mechanisms driving these differences and to assess the long-term performance of these materials in practical settings.

## Figures and Tables

**Figure 1 biomimetics-10-00235-f001:**
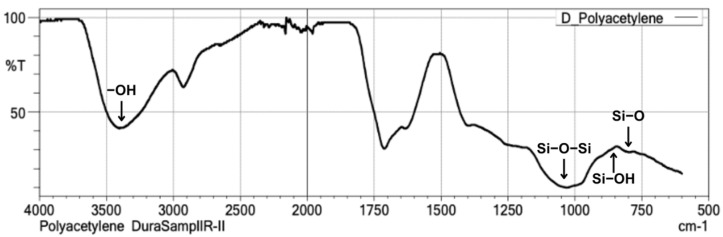
FT-IR spectra graph of silica from diatom *Thalassiosira* sp.

**Figure 2 biomimetics-10-00235-f002:**
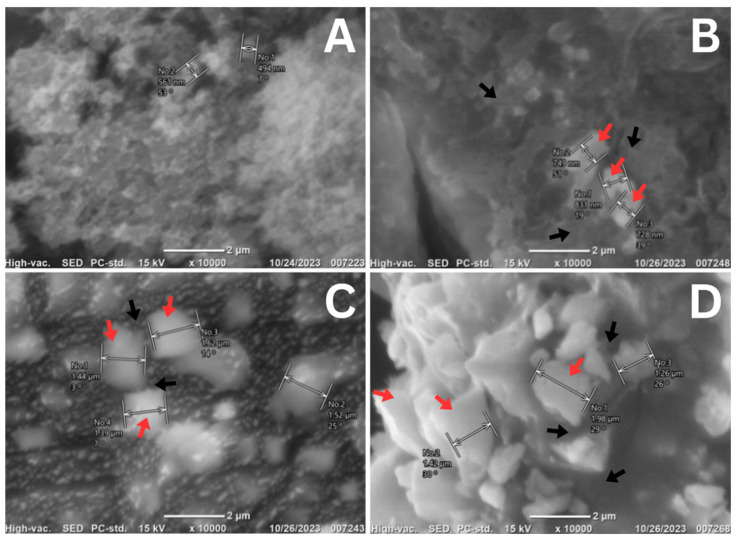
SEM results of mixing nano-silica of diatom *Thalassiosira* sp. with GIC luting at 10,000× magnification: (**A**). 0%; (**B**). 1%; (**C**). 3%; (**D**). 5%. Red arrows indicate the amorphous structure of nano-silica from *Thalassiosira* sp., while black arrows indicate the crystalline structure of Luting Glass Ionomer Cement.

**Figure 3 biomimetics-10-00235-f003:**
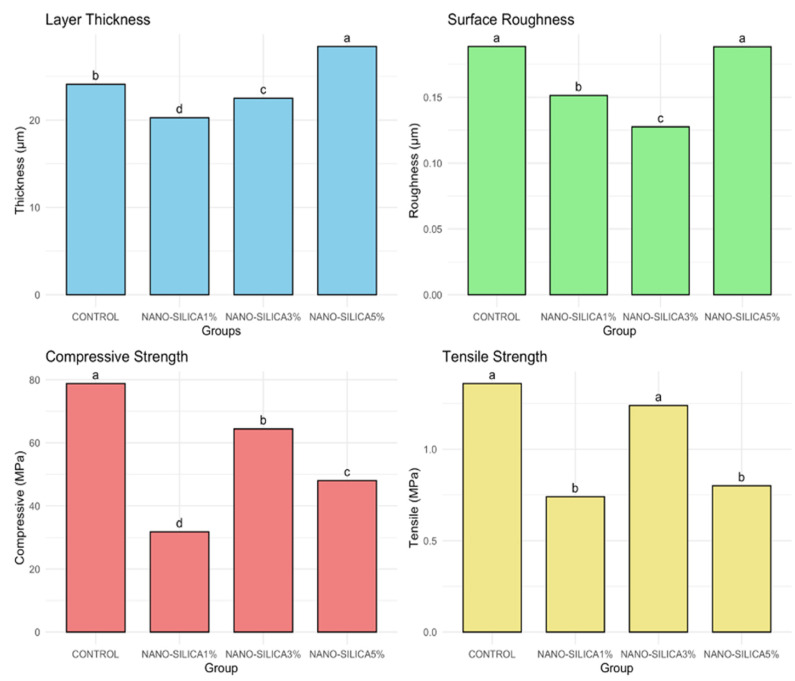
Bar graphs comparing the mean values of layer thickness, surface roughness, compressive strength, and tensile strength across different groups, with statistical significance indicated by different letters.

**Table 1 biomimetics-10-00235-t001:** Division of research groups.

Groups	Description	Name of Groups
Group 1	Glass ionomer cement (0.225 g)	CONTROL
Group 2	Glass ionomer cement (0.2227 g) + nano-silica 1% (0.0023 g)	NANO-SILICA1%
Group 3	Glass ionomer cement (0.2182 g) + nano-silica 3% (0.0068 g)	NANO-SILICA3%
Group 4	Glass ionomer cement (0.2137 g) + nano-silica 5% (0.0113 g)	NANO-SILICA5%

**Table 2 biomimetics-10-00235-t002:** Mean values and standard deviations for layer thickness across groups.

Groups	Mean (µm)	Standard Deviation	*p*-Value
CONTROL	24.11	0.68	0.000
NANO-SILICA1%	20.27	0.92	
NANO-SILICA3%	22.53	0.47	
NANO-SILICA5%	28.43	0.47	

*n*-value in each group is 3. *p*-value < 0.05 indicated significant difference using one-way ANOVA.

**Table 3 biomimetics-10-00235-t003:** Mean values and standard deviations for surface roughness across groups.

Groups	Mean (µm)	Standard Deviation	*p*-Value
CONTROL	0.189	0.009	0.000
NANO-SILICA1%	0.151	0.007	
NANO-SILICA3%	0.128	0.004	
NANO-SILICA5%	0.188	0.008	

*n*-value in each group is 3. *p*-value < 0.05 indicated significant difference using one-way ANOVA.

**Table 4 biomimetics-10-00235-t004:** Mean values and standard deviations for compressive strength across groups.

Groups	Mean (MPa)	Standard Deviation	*p*-Value
CONTROL	78.8	5.8	0.000
NANO-SILICA1%	31.8	5.6	
NANO-SILICA3%	64.4	5.4	
NANO-SILICA5%	48.0	5.5	

*n*-value in each group is 3. *p*-value < 0.05 indicated significant difference using one-way ANOVA.

**Table 5 biomimetics-10-00235-t005:** Mean values and standard deviations for tensile strength across groups.

Groups	Mean (MPa)	Standard Deviation	*p*-Value
CONTROL	1.32	0.19	0.000
NANO-SILICA1%	0.74	0.11	
NANO-SILICA3%	1.24	0.19	
NANO-SILICA5%	0.76	0.18	

*n*-value in each group is 3. *p*-value < 0.05 indicated significant difference using one-way ANOVA.

## Data Availability

Additional questions can be forwarded to the corresponding author; the article contains the original contributions made in the study.

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
