# Peer review of "Synthesis and Application of Sol-Gel-Derived Nano-Silica in Glass Ionomer Cement for Dental Cementation"

_biomimetics, 2025, doi:10.3390/biomimetics10040235_

Round 1

Reviewer 1 Report

Comments and Suggestions for Authors

The reviewer's response is provided in the attached file.

Author Response

dear reviewer, thank you all your insightful comments. we attached the file of rebuttal letter down below to respond all the comment point-by-point.

thank you once again.

Reviewer 2 Report

Comments and Suggestions for Authors

The authors present detailed and novel research, but they should clarify the following and do general editing of the text:

  1. In the Introduction part, the author should correct
  • Scientific Accuracy & Coherence (Line 53-58) "Among the numerous nanomaterials being researched, nano-silica has developed as a particularly interesting user." (Line 55) Correction: "Among various nanomaterials, nano-silica has emerged as a promising additive for GIC reinforcement."
  • The transition from GIC limitations to nano-silica benefits (Lines 53-58) needs better logical flow. Clearly explain why nano-silica is the preferred choice over other reinforcements.
  • Methodology Clarification (Lines 60-63) The sol-gel method is introduced (Lines 60-63), but the text should briefly justify why it is preferred for dental applications. A sentence explaining its advantages (e.g., better control over particle size and morphology) would enhance clarity.
  • Clinical and Future Research Relevance (Lines 73-77) The objective statement (Lines 73-77) should briefly highlight the clinical impact of the study and suggest future research directions, such as long-term biocompatibility and scalability of nano-silica integration in GICs.
  1. Comparative Interpretation (Lines 120-122) – Explain why NANO-SILICA5% led to the highest thickness and NANO-SILICA1% the lowest—was this expected based on material properties?
  2. (Lines 236-240) The author should Mention if temperature and stirring speed were controlled during pH adjustment.
  3. Titration Conditions (Lines 236-240) pls Mention if temperature and stirring speed were controlled during pH adjustment.
  4. Correction of P-Value Format (Line 179) – Change "P<0.000" to "P<0.001" for correct statistical reporting.
  5. Clarify Statistical Findings (Lines 181-185) – Explain why NANO-SILICA1% and NANO-SILICA5% had significantly lower tensile strengths—was it due to particle dispersion, bonding issues, or material brittleness?
  6. Tukey’s HSD Test Interpretation (Lines 182-183) – Since CONTROL and NANO-SILICA3% showed no statistical difference, discuss if NANO-SILICA3% is an optimal reinforcement level.
  7. In the conclusion section, the author should include a discussion on the study's limitations and future research directions.

Author Response

dear the reviewer. 

thank you for reviewing our manuscript. also, all your comments are helpful for our manuscript. we response point-by-point all your comments in the rebuttal letter down below. so, please see the attachment. 

once again, thank you
